# Improvement in the Phytochemical Content and Biological Properties of *Stevia rebaudiana* (Bertoni) Bertoni Plant Using Endophytic Fungi *Fusarium fujikuroi*

**DOI:** 10.3390/plants12051151

**Published:** 2023-03-03

**Authors:** Reema Devi, Ahmed Abdulhaq, Rachna Verma, Kiran Sharma, Dinesh Kumar, Ajay Kumar, Ashwani Tapwal, Rahul Yadav, Syam Mohan

**Affiliations:** 1School of Biological and Environmental Sciences, Shoolini University of Biotechnology and Management Sciences, Solan 173229, India; 2Unit of Medical Microbiology, Department of Medical Lab Technology, Faculty of Applied Medical Sciences, Jazan University, Jazan 45142, Saudi Arabia; 3School of Bioengineering and Food Technology, Shoolini University of Biotechnology and Business Management, Solan 173229, India; 4Himalayan Forest Research Institute, Conifer Campus, Shimla 171013, India; 5Shoolini Life Sciences, Private Limited, Solan 173229, India; 6Substance Abuse and Toxicology Research Centre, Jazan University, Jazan 45142, Saudi Arabia; 7School of Health Sciences, University of Petroleum and Energy Studies, Dehradun 248007, India; 8Center for Transdisciplinary Research, Department of Pharmacology, Saveetha Dental College, Saveetha Institute of Medical and Technical Science, Saveetha University, Chennai 600077, India

**Keywords:** antagonistic activity, antioxidant activity, DNA sequencing, endophytic fungi inoculation, UHPLC

## Abstract

This study aimed to increase the therapeutic potential of medicinal plants through inoculation with endophytic fungi. As endophytes influence medicinal plants’ biological properties, twenty fungal strains were isolated from the medicinal plant *Ocimum tenuiflorum.* Among all fungal isolates, the R2 strain showed the highest antagonistic activity towards plant pathogenic fungi *Rosellinia necatrix* and *Fusarium oxysporum*. The partial ITS region of the R2 strain was deposited in the GenBank nucleotide sequence databases under accession number ON652311 as *Fusarium fujikuroi* isolate R2 OS. To ascertain the impact of an endophytic fungus on the biological functions of medicinal plants, *Stevia rebaudiana* seeds were inoculated with *Fusarium fujikuroi* (ON652311). In the DPPH assay, the IC_50_ value of the inoculated *Stevia* plant extracts (methanol, chloroform, and positive control) was 72.082 µg/mL, 85.78 µg/mL, and 18.86 µg/mL, respectively. In the FRAP assay, the IC_50_ value of the inoculated *Stevia* extracts (methanol, chloroform extract, and positive control) was 97.064 µM Fe^2+^ equivalents, 117.662 µM Fe^2+^ equivalents, and 53.384 µM Fe^2+^ equivalents, respectively. In the extracts of the plant inoculated with endophytic fungus, rutin and syringic acid (polyphenols) concentrations were 20.8793 mg/L and 5.4389 mg/L, respectively, which were higher than in the control plant extracts. This approach can be further utilized for other medicinal plants to increase their phytochemical content and hence medicinal potential in a sustainable way.

## 1. Introduction

Endophytic fungi spend all or part of their life cycle dominating in inter- and/or intracellular systems, mainly in leaves, stems, and roots, without causing disease signs in their hosts [1]. Endophytic fungi obtain shelter and nutrients from the host plant and protection from the harsh natural environment. In return, endophytic fungi enhance the host plant’s tolerance to abiotic or biotic stresses. The coexisting interaction has produced many beneficial aspects such as the production of phytohormones, siderophores, hydrogen cyanide, phosphate-solubilizing agents, and hydrolytic enzymes, etc. [2]. In return, endophytic fungi benefit the host plant, especially in the case of phytochemical production, where they are found to mimic the production of secondary metabolites of the host, e.g., *Fusarium redolans*, which was observed to produce taxol, an anticancer drug, and was isolated from *Taxus* plant as an endophytic fungi [3]. The metabolites produced by endophytic fungi may influence the intake or redistribution of resources and accumulation of bioactive metabolites and stimulate plant growth [4,5], which can significantly increase the host plant’s vigor. However, the knowledge of the precise interactions between the endophytic fungi and their host plants is still quite limited [5]. By modifying the conditions under which medicinal plants thrive, we can comprehend and take advantage of such interactions to produce better medications [6,7]. Maize plants inoculated with endophytic fungi (*Piriformospora indica*) have boosted the activity of antioxidant enzymes and raised resistance to parasitic root fungus [8]. It was discovered that *F. fujikuroi* produces several additional beneficial secondary metabolites, i.e., methylfusarubin, gibberelline, and bikaverin, etc., indicating that it has the potential to be used in the synthesis of other compounds. Consequently, the secondary metabolites produced by *Fusarium* strains can be thought of as possible biocontrol agents against mosquitoes or nematodes that carry various diseases impacting people and plants [9].

Two medicinal plants were selected for the present study, *Ocimum* and *Stevia rebaudiana*. Among the plants known for their medicinal value, the *Ocimum* plant of the Lamiaceae family has important therapeutic potential, which has been scientifically proven [10]. *Ocimum sanctum*, also known as *Ocimum tenuiflorum* and commonly known as Basil or Tulsi, is often used in the treatment of various diseases. It offers a variety of benefits, including adaptogenic, analgesic, anti-microbial, anti-fertility, and antispasmodic characteristics [11]. The key benefits of *Ocimum* in different therapies are its safety or harmless nature, as claimed by Ayurveda, in addition to being effective, less costly, and widely available [12,13]. Thus, the *Ocimum* plant was selected for the isolation of the beneficial endophytic fungi. *Stevia rebaudiana* (Asteraceae) is a medicinal plant used to produce steviol glycosides, a type of natural sweetener that has significant economic value in food goods [14]. Diabetic patients are thought to benefit most from *Stevia*-derived chemicals as an alternate sweetener. According to statistics, stevioside-like sweet products in some countries can replace up to 30% of the sugar that is normally consumed [15]. *Stevia* compounds are widely used in industry as energizers and in beverages, as well as in medicine in vasodilators, cardiotonics, anesthetics, anti-inflammatories, and uric acid reducers. Diterpene glycosides are a type of natural sweetener derived from the *Stevia* plant [16]. This study is an attempt to enhance the medicinal potential of this plant through endophytic fungi inoculation. Natural compounds originating from fungi are crucial in the search for novel medicines [17]. Therefore, the objective of the current study was to investigate the fungal strains that live as endophytes in the medicinal plant *Ocimum tenuiflorum*. Based on the antagonistic activity, the best fungal isolate was selected from a mythoreligious and important Ayurveda plant, *Ocimum*, and inoculated in *S. rebaudiana* to assess the antioxidant potential and the phytochemicals of two groups (plants with and without endophytic fungi inoculation). Thus, the *S. rebaudiana* plant, which has its own potential role for sugar patients, may improve its current significance to some extent by inoculation with potential endophytic fungi.

## 2. Results

### 2.1. Isolation and Morphological Characteristics of Endophytic Fungi

From the leaves of *O. tenuiflorum*, 20 fungal strains *Trichoderma viride* (R1), *Fusarium fujikuroi* (R2), *Trichoderma* sp. (R3), *Alternaria alternata* (R4), *Alternaria* sp. (R5), *Alternaria* sp. (R6), *Trichoderma harzianum* (R7), *Fusarium solani* (R8), *Phoma setosa* (R9), *Penicillium citrinum* (R10), *Epicoccum nigrum* (R11), *Penicillium chrysogenum* (R12), *Absidia* sp. (R13), *Aspergillus versicolor* (R14), *Pythium* sp. (R15), *Pythium* sp. (R16), *Absidia ramosa* (R17), *Mortierella* sp. (R18), *Rhizopus oryzae* (R19), and *Rhizopus nigricans* (R20) were isolated and identified. The R2 strain was selected based on antagonistic activity and identified as *F. fujikuroi* through sequence analyses of the fungal 18S ITS region. The isolated *F. fujikuroi* sequences were very like those in the NCBI’s GenBank database. The Fusarium isolate’s partial ITS regions of the rRNA sequences reported in this study have been deposited in the NCBI gene nucleotide sequence databases (http://www.ncbi.nlm.nih.gov; dated: 2 June 2022) with accession number ON652311 (*Fusarium fujikuroi* isolate R2 OS). The fungal isolates were classified as members of ascomycota (R1-R14), zygomycota (R15-R17), or oomycota (R18-R20).

### 2.2. Antagonistic Impact of Endophytic Fungi against Plant Pathogenic Fungi

The antagonistic activity testing was performed on all the isolated endophytic fungi against two plant pathogens, *Rosellinia necatrix* and *Fusarium oxysporum*. As shown in Figure 1, antagonistic studies on endophytic fungal isolates revealed a significant reduction in pathogen development in terms of radial diameter. Out of the tested twenty fungal isolates, the R2 strain of endophytic fungi showed the highest level of antagonism (Table 1), followed by isolates R2, R11, R12, R1, and R15, with the remaining isolates showing very little activity against both pathogenic fungi.

Fungal strains were observed to be potentially antagonistic to plant pathogenic fungi in this screening study, with growth inhibition ranging from 26.98% to 42.85% in the case of *R*. *necatrix* and 32.13% to 45.82% in *F. oxysporum.* Table 1 shows that the highest antagonism of the R2 strain against both pathogenic fungi was 42.82% and 45.82% against *R*. *necatrix* and *F. oxysporum*, respectively.

### 2.3. Aligned Sequence Data of Sample—Fusarium Fujikuroi (551 bp)

The DNA sequence of the antagonistic fungi *F*. *fujikuroi* was determined, with accession number ON652311 (isolate R2 OS). The following is the 18S-ITS sequence of the isolated fungi:

TTCCGTAGGGTGAACCTGCGGAGGGATCATTACCGAGTTTACAACTCCCAAACCCCTGTGAACATACCAATTGTTGCCTCGGCGGATCAGCCCGCTCCCGGTAAAACGGGACGGCCCGCCAGAGGACCCCTAAACTCTGTTTCTATATGTAACTTCTGAGTAAAACCATAAATAAATCAAAACTTTCAACAACGGATCTCTTGGTTCTGGCATCGATGAAGAACGCAGCAAAATGCGATAAGTAATGTGAATTGCAGAATTCAGTGAATCATCGAATCTTTGAACGCACATTGCGCCCGCCAGTATTCTGGCGGGCATGCCTGTTCGAGCGTCATTTCAACCCTCAAGCCCAGCTTGGTGTTGGGACTCGCGAGTCAAATCGCGTTCCCCAAATTGATTGGCGGTCACGTCGAGCTTCCATAGCGTAGTAGTAAAACCCTCGTTACTGGTAATCGTCGCGGCCACGCCGTTAAACCCCAACTTCTGAATGATGACCTCGGATCAGGTAGGAATACCCGCTGAACTTAAGCATATCAATAAGCGGGAGGGAA

#### ITS1-5.8S rRNA-ITS2 Region Sequences Analyzed Phylogenetically

The ITS1-5.8S rRNA-ITS2 region of *Fusarium* isolate R2 OS proved to be extremely similar to *F. fujikuroi* in terms of data sequencing, with a total score of 1147 and 99.64% similarity. Multiple sequences from the GenBank database representing *Fusarium* species were used to construct a phylogenetic tree to evaluate the position of each strain in a phylogenetic analysis. The phylogenetic tree as given in Figure 2 revealed that *Fusarium* isolate (R2 OS) and *F. fujikuroi* endophytic fungi belonged to the same clade. As a result, *Fusarium* isolate R2 OS was named as *F. fujikuroi*.

### 2.4. Antioxidant Assay

The antioxidant potential of the plant extract after inoculation with endophytic fungi was determined using DPPH (2,2-diphenyl-1-picrylhydrazyl) and FRAP (Ferric reducing antioxidant power) assays.

#### 2.4.1. DPPH Assay

Both *Stevia* extracts (methanol and chloroform) reduced the oxidation activity of DPPH radicals because of the radical scavenging ability. The DPPH assay results revealed that when the extract concentration increased (20–100 µg/mL), the percentage inhibition of free radicals also increased. The greatest percentage inhibition was found to be 61.58 ± 0.51% in the methanol extract, while the chloroform extract demonstrated the least amount of inhibition at a concentration of 100 µg/mL (55.20 ± 0.81%). In the case of the control plants (non-inoculated), the DPPH radical scavenging activities of the extracts (methanol and chloroform) ranged from 19.46% to 41.94% and 17.16 % to 40.45%, respectively. The IC_50_ values of the control plant extracts (methanol and chloroform) were 129.676 µg/mL and 135.603 µg/mL, respectively, and in the case of the inoculated plant extracts (methanol, chloroform, and positive control), the IC_50_ values were 72.082 µg/mL, 85.78 µg/mL, and 18.86 µg/mL, respectively (Figure 3).

#### 2.4.2. FRAP Assay

The plant extracts were tested for their oxidation-reduction ability by converting Fe^3+^ to Fe^2+^. Depending on the concentration and reduction power of the molecules present in the plant extracts, the test solution’s color changed from yellow to various shades of green and blue. Similar to the DPPH assay, plant extracts of methanol and chloroform were shown to have positive results for reducing power assay. Different concentrations of plant extracts (20–100 µg/mL) were used for the assessment of percent inhibition in the FRAP assay. At 100 µg/mL, the methanol extract (50.18 ± 0.40 M/mL FeSO_4_ equivalents) showed higher percentage inhibition than the chloroform extract (45.15 ± 0.56 M/mL FeSO_4_ equivalents). The antioxidant activity of the *S. rebaudiana* control plant extracts (methanol and chloroform) ranged from 19.14 to 38.09 M/mL and 15.05 to 35.47 M/mL, respectively. The IC_50_ value of the plant extracts (methanol and chloroform) in the control was 149.639 M Fe^2+^ equivalents and 154.788 M Fe^2+^ equivalents, respectively. The antioxidant potential of the *Stevia* extracts (with inoculated plants), viz., methanol and chloroform, was found to be 97.064 M Fe^2+^ equivalents and 117.662 M Fe^2+^ equivalents, respectively (Figure 4).

### 2.5. UHPLC (Ultra-High-Performance Liquid Chromatography)

Rutin, quercetin, gallic acid, caffeic acid, and syringic acid (polyphenols) present in the plant extracts (methanol and chloroform) might be responsible for different bioactivities and were assessed for their presence in the plant extracts by UHPLC analysis. Various polyphenols were seen under the curve with different retention times. In the methanolic extract, rutin was observed with the highest concentration, i.e., 20.8793 mg/L, at 8.593 retention time (RT). At 350 nm, the absorbance of both quercetin and rutin was also measured, and it was observed that rutin was highest in *Stevia* methanolic plant extract at 8.593 RT when compared with the control plant extracts, i.e., methanol and chloroform (Figure 5). In contrast, UHPLC analysis of chloroform extracts of *S. rebaudiana* revealed the presence of the least amount of polyphenols when compared to methanolic extract of *S. rebaudiana*, as shown and summarized in Table 2 and Figure 6. Gallic acid, caffeic acid, syringic acid, p-coumaric acid, and salycylic acid standards were compared to the *Stevia* plant extracts to observe polyphenols such as gallic acid, caffeic acid, syringic acid, p-coumaric acid, and salycylic acid. It was observed that syringic acid in methanolic extract had a concentration of 5.4389 mg/L at 3.270 RT (Figure 7 and Table 2). Figure 8 shows the UHPLC results of polyphenols observed with chloroform plant extracts of *S. rebaudiana*.

## 3. Materials and Methodology

### 3.1. Sample Collection

In separate sterile polythene bags, fresh, healthy, and mature plants of *O. tenuiflorum* were collected from the campus of Shoolini University, Solan district, Himachal Pradesh, India. The sampling site in Solan district lies in the latitudinal range of 30°44′53″ to 31°22′1″ N and longitudinal range of 76°36′10″ to 76°15′14″ E, with wide altitudinal ranges. Endophytic fungi were isolated from the fresh plants and used for the research. The *Stevia* and *Ocimum* plant samples were used for preparing the herbarium which was authenticated by BSI (Botanical Survey of India) in Nauni, Solan, Himachal Pradesh (Accession Nos. 00043 and 00044).

### 3.2. Endophytic Fungi Isolation and Cultivation

Cui et al.’s [18] method was adopted for isolating endophytic fungi from healthy *O. tenuiflorum* plant leaves. The sample was placed in a sterile bag, and within 24 h of collection the fungal endophytes were isolated. The plant samples were washed twice with autoclaved distilled water and three times with regular distilled water. Following a sterile distilled water rinse, the surface of each treatment sample was cleaned and disinfected using 70% ethanol for 60 s, 1% sodium hypochlorite (NaOCl) again for 60 s, and 70% ethanol for 30 s. To get rid of extra moisture, the treated samples were placed on sterilized blotting paper. Surface-sterilized samples were prepared with endophytic fungi cut into 6 to 6 mm long pieces and mounted on 90 mm Petri dishes with water agar medium (2% agar by volume). To stop bacterial growth, streptomycin sulphate (200 µg/mL) was added to the media. Until mycelium from the inoculated plant samples appeared, the Petri dishes were cultured for 3–12 days at 28 °C. Inoculated samples produced hyphal growth which was collected and put on potato dextrose agar media before being cultured at 28 °C to obtain a pure culture. Each isolate was grown on PDA agar and stored at 4 °C for later use. Morphological characteristics such as mycelium color, colony morphology, and reverse media color were used to identify endophytic fungi.

### 3.3. Fungal Isolates for Antagonistic Activity

The antagonistic effect of fungal endophytes on plant pathogenic fungi was investigated using the dual-culture technique [19]. On sterile PDA plates, the pathogen and antagonists were grown separately for 5 days. On sterile PDA plates, the antagonists and the pathogen were grown separately for five days. A 5 mm endophyte culture was positioned in one direction, 1 cm away from the PDA-media-containing Petri plate’s edge, and a test pathogen of the same size was obtained by placing it in another direction. Plant pathogenic fungi *R. necatrix* (accession no. ON652311) and *F. oxysporum* (accession no. SR266-9) were obtained from the School of Applied Sciences and Biotechnology in Shoolini University, Solan. Using the pathogen alone without the antagonist (endophytes) as control, the experiment was carried out in triplicate. The pathogen mycelia totally covered the control plates after 12 h in the dark and 12 h in the light at 25 ± 2 °C. To ensure equal growth opportunities, antagonist and pathogen paired cultures were placed at equal distances from the periphery. Following incubation, the radial growth of the control and treatment plates was measured, the radial growth of colonies was measured after 5 days, and the formula used to determine the percentage was:I = C − T × 100\C
where I = mycelial growth inhibitor;

T = radial mycelial development of the antagonistic fungus toward the pathogen (T);

C = radial mycelial development of that on a control plate.

### 3.4. DNA Extraction and PCR Amplification

#### 3.4.1. DNA Extractions

The isolated endophytic fungi from *O. tenuiflorum* were further subjected to molecular characterization. The CTAB method was used to extract the genomic DNA of the fungi. We added 100 mg of the sample and 1 mL of the extraction buffer in a crusher and pestle to homogenize it. To extract DNA, the homogenate was put in a 2 mL microfuge tube. The tubes were filled with equal amounts of phenol, chloroform, and isoamyl alcohol (25:24:1) and gently shaken to properly mix up the reagents. At room temperature, the tubes were centrifuged at 14,000 rpm for 15 min. The upper aqueous phase thus obtained was subjected to centrifugation at 14,000 rpm for 10 min and transferred to a fresh new tube. The DNA was precipitated from the solution by adding 0.1 volume of 3 M sodium acetate pH 7.0 and 0.7 volumes of isopropanol. Following a 15 min incubation period at ambient temperature, the tubes were centrifuged at 4 °C for 15 min at 14,000 rpm. The DNA pellet was thoroughly cleaned twice with 70% ethanol, quickly rinsed with 100% ethanol, and then left to dry naturally. To remove RNA, the DNA was treated with 5 μL of DNAse-free RNAse A (10 mg/mL) after dispersing in TE (Tris-Cl 10 mM, pH 8.0, EDTA 1 mM) [20]. A certain amount (50 μL) of total reaction volume, 10 pM of each primer, and a total of 127 ng of extracted DNA were used for amplification. High-fidelity PCR polymerase was used to amplify the 700 bp ITS fragment. The effectiveness of PCR amplification depends on using the proper primary pair and the appropriate annealing temperature for fungus. The main primer pairs employed in this investigation were ITS1 (forward), which served as a fungus-specific primer, and ITS4, which served as a universal primer. The annealing temperatures for primers were 57 °C and 53 °C, and the other details of the primers are mentioned in Table 3.

#### 3.4.2. Quantity and Quality Determination

The amount of extracted gDNA was calculated using a Thermo Scientific Nano Drop 1000 spectrophotometer with measuring absorbance at 260 nm. The quality of the extracted gDNA and its eligibility for eventual use in Random Amplified Polymorphic DNA (RAPD) was used to assess the genomic relationship. The polycistronic gene and multi-copy genetic amplification were determined by putting the isolated gDNA through 0.8% agarose gel electrophoresis. We used 100 ng of genomic DNA, 2.5 mL of 10 mM dNTPs mix, 2 μL of random decamer oligo template OPA-1 (5′-CAGGCCCTTC-3′), and 1 µL (5 U/µL) of dNTPs in the PCR reactions for the RAPD evaluation in a 25 μL reaction volume (Sigma-Aldrich). An Eppendorf master cycler was used for amplification (Eppendorf, Hamburg). After three minutes of initial denaturation at 94 °C, the process went through 30 cycles of one minute of denaturation, one minute of annealing at 50 °C, two minutes of extension at 72 °C, and then extensions at 72 °C for ten minutes [21].

#### 3.4.3. Sequence Alignment and Phylogenetic Analysis

The MEGA 11 software’s ClustlW algorithm was used to align the resulting DNA sequences. The BLASTn tool was used to perform a homology search on the NCBI GenBank database (https://blast.ncbi.nlm.nih.gov/Blast.cgi; dated: 2 June 2022), and the fungal isolates were identified using the percent homology scores. The phylogenetic tree was produced with MEGA version 11 using the neighbor-joining algorithm, the position estimation technique, and bootstrapping analyses for 1000 replicates. Partial ITS regions of the rRNA sequences of the *Fusarium* isolate reported in this study were deposited in the GenBank nucleotide sequence databases (http://www.ncbi.nlm.nih.gov; dated: 2 June 2022) with accession number ON652311 (*Fusarium fujikuroi* R2 OS).

### 3.5. Preparation of Fungal Inoculum

Minor modifications were made to the methods in [22] for the preparation of the fungal inoculum. Two or three pieces of the culture were inoculated into a 250 mL Erlenmeyer flask containing 100 mL potato dextrose broth, and the flask was then allowed to develop a pure culture for 15 days at 25 °C and 120 rpm. To detach the mycelia and filtrate from the broth culture, sterile cheesecloth was used as a filter.

#### 3.5.1. Inoculation Method

*Stevia* seeds were surface-sterilized with 2% NaOCl (sodium hypochlorite) solution for 1 min before being washed with sterile water to remove traces of NaOCl. For the sterile soil assay, 15 × 10 cm plastic pots were used, and a potting mixture weighing 1.5 kg was placed inside each pot. A 10% (*w*/*v*) carboxyl methyl cellulose, or CMC, adhesive was used to wrap the *Stevia* seeds with an endophytic fungus cell suspension that had an antagonistic effect on pathogenic microbes. As sterilized air was being blown over them, the coated seeds were dried by air for a couple of hours. The coated seeds were then planted in 1.5 cm pots, with an average of four germinations occurring in each pot. The following treatments were studied: (i) control: no endophytic fungi inoculation (Figure 9A); (ii) endophytic fungi inoculation in *S. rebaudiana* plants under greenhouse conditions (Figure 9B). The pots were placed in a random pattern, and after two months of inoculation the plant leaves were collected for further examination [23].

#### 3.5.2. Plant Extracts Preparation

With a few minor modifications, the Behera et al. [24] method was used to prepare the plant extracts. For this purpose, the effective plant part, i.e., leaves, were shade-dried for 10–15 days, grounded to powder form, and, using a conical flask, placed in a rotatory orbital shaker at 40 °C for 48 h. A total of 10 g of powder was extracted by adding 100 mL of polar solvent (methanol) and 100 mL of non-polar solvent (chloroform). The supernatant was then filtered using Whatman filter paper, dried in a water bath at 40 °C, and placed in the refrigerator for further use.

### 3.6. Antioxidant Activity

#### 3.6.1. 2,2-Diphenyl-1-Picrylhydrazyl (DPPH) Radical Scavenging Activity

The Katalinic [25] method was used to assess the extract’s scavenging potential of DPPH. The activity was measured by preparing plant extract in methanol at different concentrations (5 µg/mL, 10 µg/mL, 20 µg/mL, and 40 µg/mL). Ascorbic acid was used as standard at a concentration comparable to that of the sample. A total of 100 µL of the extract was combined with 900 µL of DPPH methanolic solution (0.004%). The combination was kept at room temperature for 30 min. Following a 30 min incubation, the purple color disappearance was assessed with a UV/visible spectrophotometer.

We calculated the radical scavenging activity of DPPH using the following formula:$$\mathrm{Inhibition}\;\%=\frac{\left(\mathrm{Control}-\mathrm{Test}\right)\times 100}{\mathrm{Control}}$$

#### 3.6.2. FRAP Assay

The FRAP activity was determined using the Benzie and Strain [26] methodology. To make the FRAP reagent, in a 10:1:1 ratio, 10.0 mM TPTZ (tripyridyltriazine) solution, 20.0 mM FeCl_3_·6H_2_O solution, and 300 mM sodium acetate buffer (pH 3.6) were combined. In order to analyze samples with different concentrations (20 µg/mL, 40 µg/mL, 60 µg/mL, and 80 µg/mL), three milliliters of FRAP reagent were applied. The reaction mixtures were maintained at 37 °C for 30 min. The absorbance was calculated at 593 nm. A fresh working solution of FeSO_4_ was utilized for calibration. Depending on how well the sample could begin decreasing ferrous ions, the antioxidant capacity was calculated and demonstrated as FeSO_4_ equivalents per gram of sample.

### 3.7. UHPLC

A Thermo-fisher scientific Dionex Ultimate 3000 series equipment (Thermo-fisher Scientific, USA) chromatographic system was used for all HPLC analyses (at Food Testing lab, Shoolini University, Solan, Himachal Pradesh), consisting of a vacuum degasser, quaternary pump, autosampler with thermostat, column compartment thermostat, and photodiode array detector (DAD). The system was coupled with Zorbax Eclipse C18 (4.6 mm × 250 mm, 5 µm) column (Agilent Technologies, Santa Clara, CA, USA). The mobile phase involved two components for the analysis of polyphenol (rutin and quercetin) content, combining HPLC water with pH 2.5 (solvent A) and HPLC-grade acetonitrile (solvent B) at a flow rate of 0.07 mL/min, column oven temp. 25 °C, and UV 350 nm in gradient mode. The compounds were detected using certified reference materials (CRMs) (Sigma Aldrich). The 20 uL sample was injected into the injector with the help of an autosampler. With the help of chromeleon 7 software in UHPLC, the compounds were identified [27]. For the analysis of polyphenols (gallic acid, caffeic acid, and syringic acid) content, the mobile phase was 1% acetic acid (A): acetonitrile (B) (70:30) in isocratic mode at flow rate 1 mL/min, column oven temp. 30 °C, and UV 280 nm. Polyphenols were detected using CRMs (Sigma Aldrich) with the help of chromeleon 7 software in UHPLC.

### 3.8. Statistical Analysis

The findings of the analyzed data were assessed by analysis of variance (one-way ANOVA) and are presented as mean ± standard deviation, with values calculated in triplicate. To identify significant differences, the Bonferroni multiple comparison test was applied. Graph Pad Prism software was utilized for statistical analysis.

## 4. Discussion

Similar rates of inhibition were observed against phytopathogenic fungi with the EF, and the R2 strain of the endophytic fungi showed the highest level of antagonism among the twenty fungal isolates examined. Morphological characteristics such as mycelium color, colony morphology, and reverse media color were used to identify the endophytic fungi [28,29,30]. Endophytes produce several chemicals that successfully suppress the growth of pathogenic fungi. Specific metabolic interactions can cause the production of secondary metabolites, and these metabolites may correspond to the fungal species’ ecological niche and taxon [31].

Some elicitors, such as glycoprotein, polysaccharides, and lipopolysaccharides, activate plant defense mechanisms and increase the secretion of secondary metabolites, effectively preventing pathogen attack [32]. The chemicals cannot be produced by many endophytic strains on their own. In fact, induced metabolism assists in metabolizing the byproduct of the other processes and boosts the production of metabolites [33]. Significant, physiologically active secondary metabolites called polyphenols are involved in the regulation of plant growth, development, and stress resistance. It has been demonstrated that a variety of biological or abiotic stimuli significantly enhance the number of phenolic compounds in plants [34]. The amount of polyphenols and alkaloids present in the roots and stems, as well as the activity of polyphenol oxidases in the stems and leaves and the activity of the acid phosphatase enzyme in the leaves, are all significantly affected by the inoculation of endophytic fungus [35]. These phenolic chemicals are also necessary for lowering oxidative stress as they are involved in the detoxification of reactive oxygen species (ROS) [36].

An endophytic fungi *Piriformospora indica* affected the gene expression of the plant *Glycine max*, thereby enhancing iron transport, lignin biosynthesis, hormone signaling, nutrient acquisition, and the biosynthesis of phenylpropanoids, flavonols, siderophores, and flavonoids. A total of 238 genes were involved in encoding the heat shock protein, and several other abiotic-stress-related defence responses [37]. There are several different bioactive secondary metabolites and phytohormones found in endophytic fungi [38]. Endophytic fungi may also be involved in biosynthesizing important secondary metabolites that are currently utilized commercially, including antibiotics, anticarcinogenics, cytotoxics, insecticides, and allelopathic compounds [39]. Additionally, endophytic fungi produce lignins, phenols and phenolic acids, flavonoids, saponins, alkaloids, terpenoids, polyketides, phenylpropanoids, and saponins when they are present in plants. This results in increased plant growth and development and pathogen resistance [38].

Secondary metabolism considerably changes in the symbionts after endophyte interaction with plants. According to Ludwig-Müller et al. [40], these modifications may result from (a) the endophyte impact on altering the host’s metabolism, (b) the host inducing endophyte metabolism, (c) the host and endophyte sharing some parts of a particular pathway, (d) the host metabolizing endophyte products, and (e) the endophyte’s capacity to degrade the host’s secondary metabolites. Endogenous fungal components as elicitors have several advantages. (i) Fungi can constantly interact with host cells, release metabolites, and expand along with the host’s growth. This can continuously activate the host’s defensive mechanism. (ii) Without showing obvious signs of infection, fungi can form a long-lasting symbiotic connection with the host [41]. The relationship between grass and *Epichloe* is a prime example of plant–endophyte mutualism [42]. The fungus secretes enzymes that weaken the cell walls of the epidermis to allow fungal proliferation into the cortical region. Endophytes in *Echinacea* plants produced indole-3-acetic acid, which altered the physiology and function of the roots [43]. *Rhizopus oryzae*, an endophytic fungus, was inoculated with *Helianthus annuus* and *Glycine max* and was shown to have significantly lower amounts of abscisic acid (ABA) and higher levels of proline, phenolics, flavonoids, ascorbic acid oxidase, and catalase [44]. The *S. rebaudiana* plant showed higher antioxidant potential after inoculation with *F. fujikuroi,* as compared with the control plant (*S. rebaudiana*) in the current study. Similarly, studies by Bagheri et al. [45]; Guler et al. [46]; and Hamayun et al. [47] observed that *Piriformospora indica* inoculated in *Oryza sativa, Trichoderma atroviride* inoculated in *Zea mays* under drought stress, and *Gliocladium cibotii* inoculated in *Glycine max* and *Helianthus annuus* under heat stress increased the antioxidant potential of the host plants. In our investigation, the methanol extract showed the highest percentage inhibition (61.58 ± 0.51%), whereas the chloroform extract showed the lowest percentage inhibition at 100 µg/mL (55.20 ± 0.81%) concentration, and the antioxidant potential was found to be at higher levels when compared with the control condition. Phenolic and flavonoid compounds are mainly responsible for the antioxidant property [48]. One of the beneficial flavonoid compounds found in high concentrations in plants is rutin [49]. Peres et al. [50] and Cardona et al. [51] found in their studies that rutin participated in antioxidant activity. In the UHPLC analysis, rutin in *Stevia* plants was 20.8793 mg/L which was higher thanthe control plants (not inoculated with any fungi). The *Fusarium oxysporum* species complex contains several strains which are commonly found in the soil. *Bacopa monniera* was co-cultivated with *Piriformospora indica,* and after that the plant was found to have increased biomass and antioxidant activity [52]. Recent research has demonstrated that endophytic fungi produce exopolysaccharides necessary for plant–endophyte interactions, and these biopolymers are distinguished by morphological structure and have potential antioxidant activity [53,54].

Phytochemicals are largely linked to the phenylpropanoid/polyphenol metabolism in plants [55]. Since many plant antioxidants are phenol derivatives, the inoculation’s minimal effect on total polyphenols suggests that the microbe–host symbiosis has a more focused physiological effect than a general increase in total polyphenol biosynthesis [55]. Phenolic buildup is facilitated by endogenous fungal components as elicitors. Additional signaling channels can also be involved, such as ROS signaling pathways, ion fluxes, and Ca^2+^ and Jasmonic acid pathways, which all influence phenolic accumulation [56,57]. According to Cappellari et al. [58], growth-promoting rhizobacteria treatments boosted *Mentha piperita*’s PAL activity, which led to the buildup of phenols. Thus, the conclusion can be made that endophytic fungi assist the plant in the formation and synthesis of secondary compounds, which enhances its antioxidant potential in mutualistic plant–fungus interactions.

Due to their wide range of advantageous effects on human health, phenolic compounds have recently received a lot of attention. The best-described property of practically all phenolic compound groups is their potential to eliminate free radicals and inhibit other oxidation reactions [59]. Gallic acid and quercetin were not detected in *S. rebaudiana* at the concentration of mg/100 g DW [60]. UHPLC analysis of *S. rebaudiana* ethanolic extracts revealed the presence of the least amount of quercetin, with 1.9 mg/L at 3.680 RT, and rutin, with 1.3 mg/L at 6.004 RT [61]. Caffeic acid and p-coumaric acid (polyphenols) were not found in the UHPLC study of *S. rebaudiana* by Oliveira et al. [60]. In our investigation, methanolic extract from *S. rebaudiana* plant inoculated with endophytic fungi had 0.46 mg/L of caffeic acid, as observed by UHPLC analysis.

## 5. Conclusions and Recommendations

Fungal isolates from the medicinal plant *O. tenuiflorum* have shown antagonistic activity towards plant pathogenic fungi, viz., *R. necatrix* (SR266-9) and *F. oxysporum* (HG964402.1). The isolate was inoculated on the *S. rebaudiana* plant, and the extracts of the same plant showed higher antioxidant potential as compared with the control plant (*S. rebaudiana*). In the UHPLC analysis, rutin and syringic acid (polyphenols) concentrations in *Stevia* plants inoculated with endophytic fungi were found to be comparatively higher than the control plants. Endophytic fungi’s impact on plant yield and quality should be identified and prioritized in all economic and medicinal plants for future use. This also serves as a sustainable way to increase the therapeutic potential of medicinal and aromatic plants, as fungal inoculation increases the phytochemical compounds in the host plants. Future research should focus on separating bioactive compounds from endophytic fungi to purify those molecules for use in the development of potential drugs. Identifying responsible biosynthetic genes for the numerous secondary metabolites from endophytic fungi opens the opportunity to explore the genetic potential of producer strains to discover novel secondary metabolites and enhance secondary metabolite production by metabolic engineering, resulting in novel and more affordable medicines and food additives.

## Figures and Tables

**Figure 1 plants-12-01151-f001:**
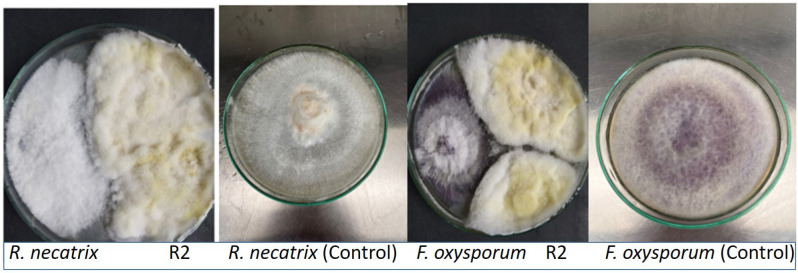
Antagonism shown by R2 strain of endophytic fungi isolated from *O. tenuiflorum* against the pathogenic fungi *Rosellina necatrix* and *Fusarium oxysporum*.

**Figure 2 plants-12-01151-f002:**
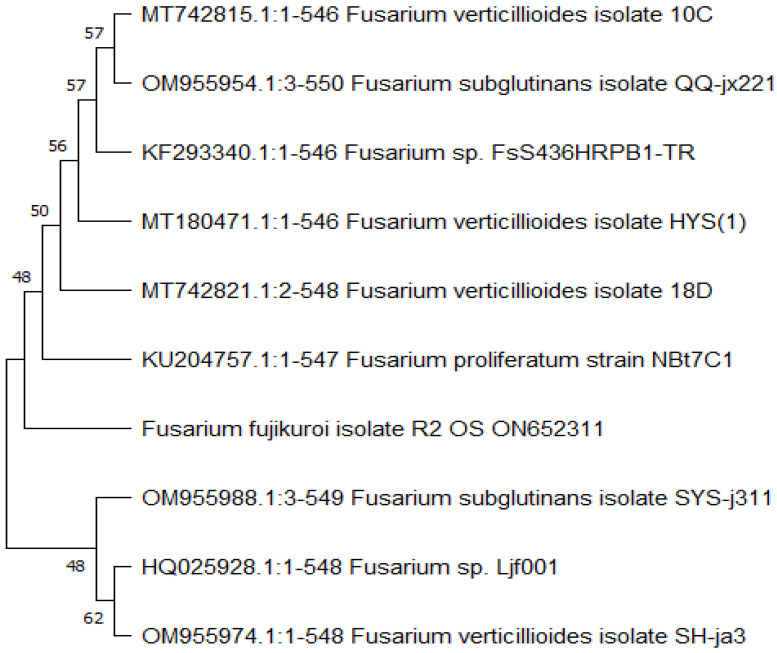
The phylogenetic relationship between isolate *Fusarium fujikuroi* and other related species.

**Figure 3 plants-12-01151-f003:**
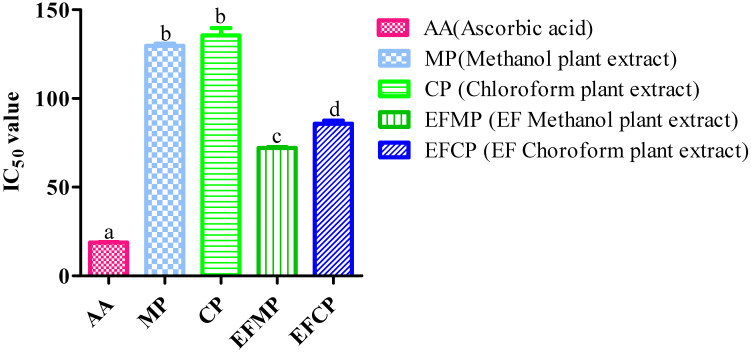
Antioxidant activity (DPPH assay) of different extracts of *S. rebaudiana.* The values are calculated as mean ± standard deviation of the three replications (n = 3), and different letters are used for significant results (*p* < 0.05).

**Figure 4 plants-12-01151-f004:**
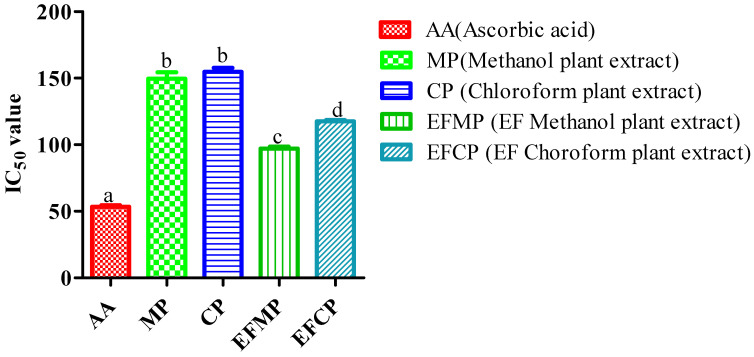
Antioxidant activity (FRAP assay) of different extracts of *S. rebaudiana.* The results are presented as mean ± standard deviation where the tests were performed in triplicate, and different letters are used for significant results (*p* < 0.05).

**Figure 5 plants-12-01151-f005:**
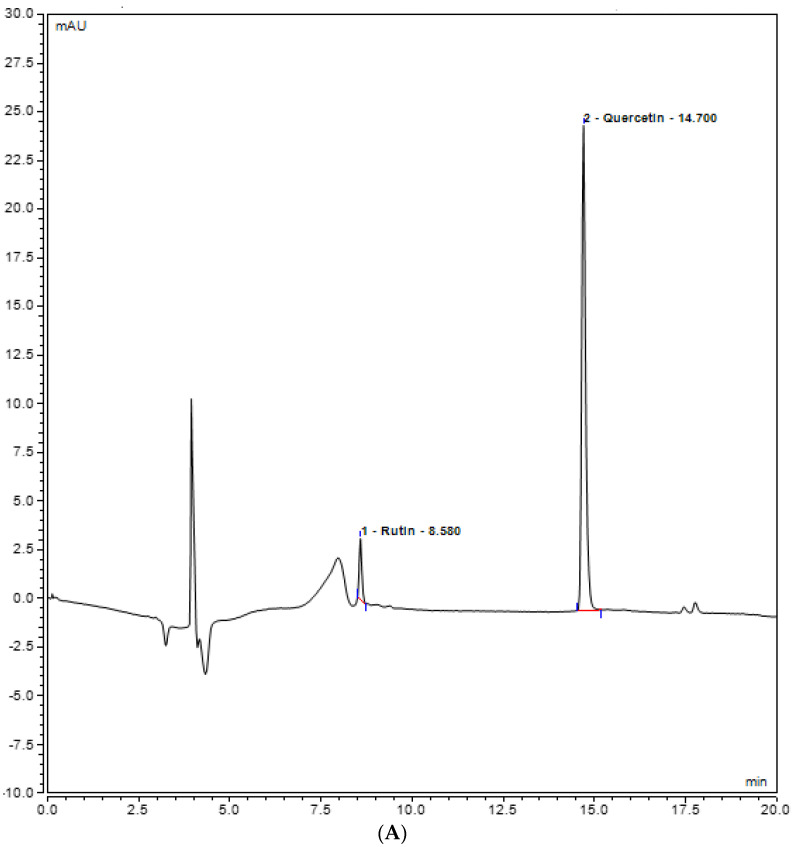
UHPLC chromatogram of polyphenols (Rutin and Quercetin) (**A**) methanol plant extract of inoculated *S. rebaudiana* with endophytic fungi, and (**B**) methanol plant extract of non−inoculated *S. rebaudiana* plant.

**Figure 6 plants-12-01151-f006:**
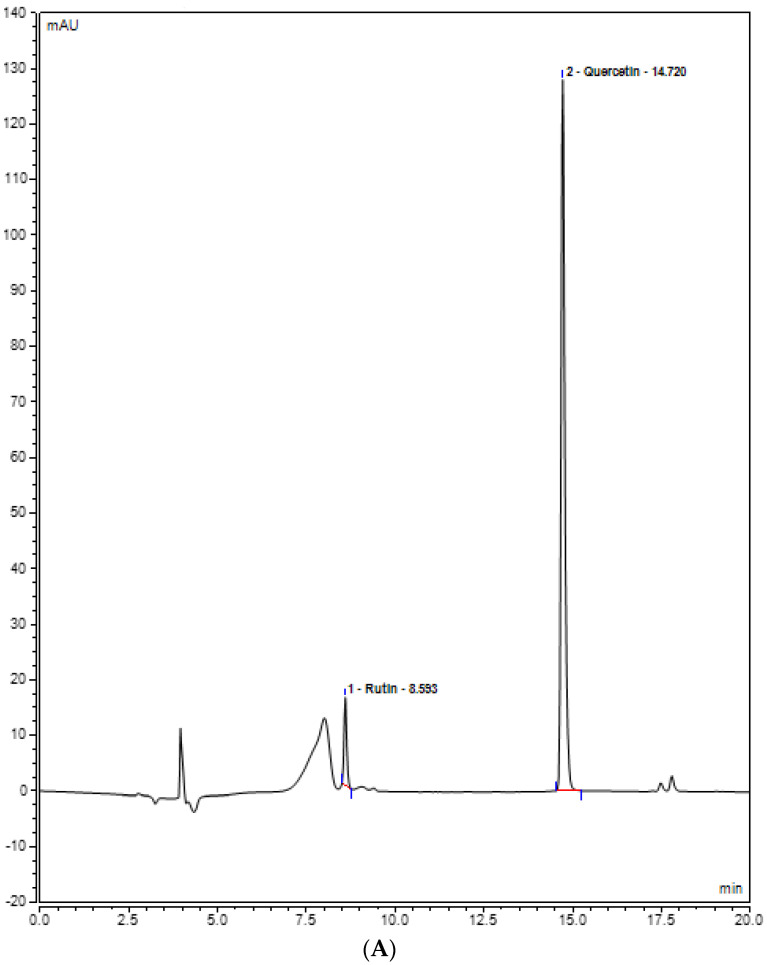
UHPLC chromatogram of polyphenols (Rutin and Quercetin) (**A**) chloroform plant extract of inoculated *S. rebaudiana* with endophytic fungi, and (**B**) chloroform plant extract of non−inoculated *S. rebaudiana* plant.

**Figure 7 plants-12-01151-f007:**
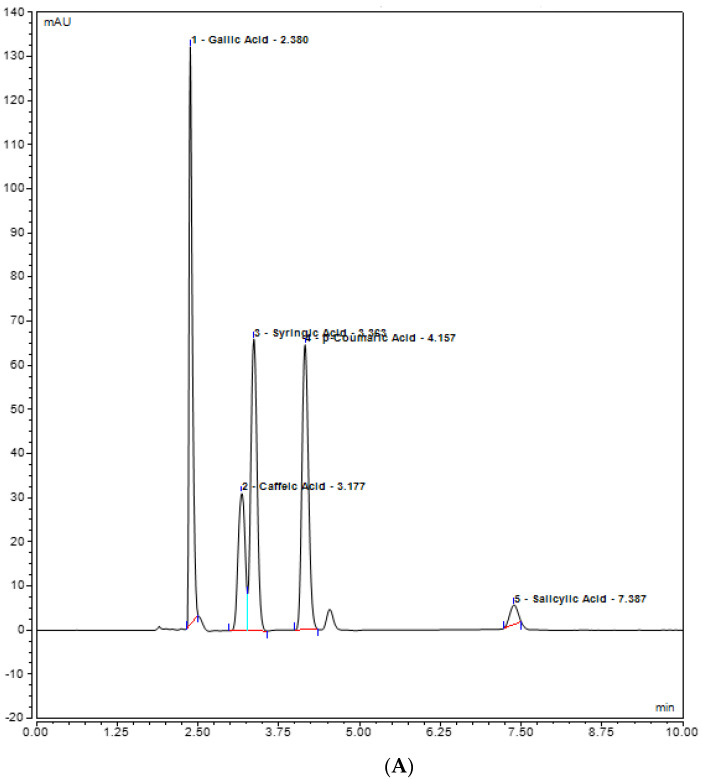
UHPLC chromatogram of polyphenols (Gallic acid, Caffeic acid, Syringic acid, p-coumaric acid, Salicylic acid) (**A**) methanol plant extract of inoculated *S. rebaudiana* with endophytic fungi, and (**B**) methanol plant extract of non−inoculated *S. rebaudiana* plant.

**Figure 8 plants-12-01151-f008:**
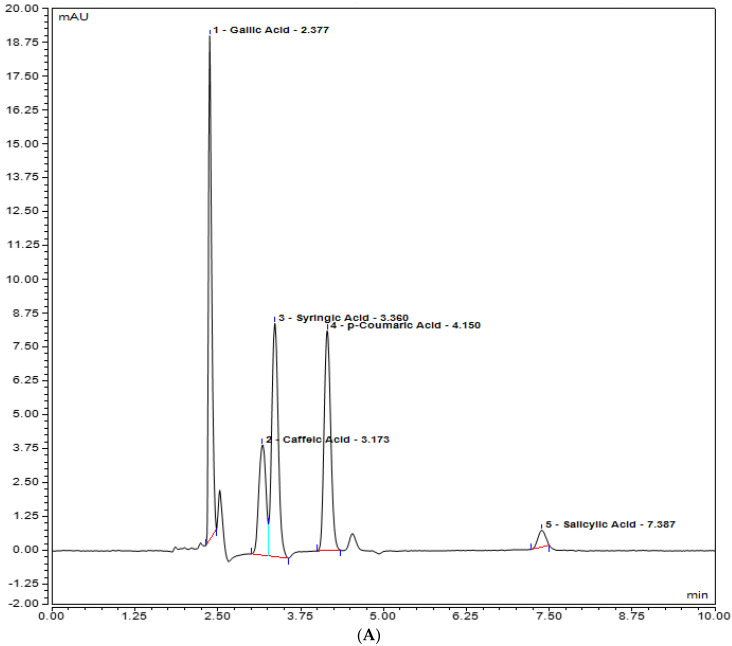
UHPLC chromatogram of polyphenols (Gallic acid, Caffeic acid, Syringic acid, p-coumaric acid, Salicylic acid) (**A**) chloroform plant extract of inoculated *S. rebaudiana* with endophytic fungi, and (**B**) chloroform plant extract of non−inoculated *S. rebaudiana* plant.

**Figure 9 plants-12-01151-f009:**
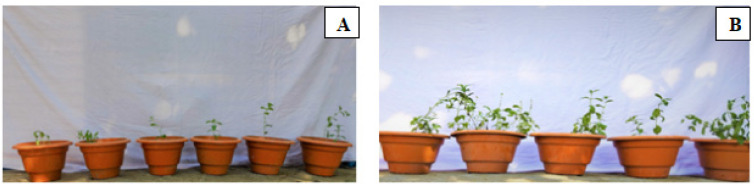
(**A**) *Stevia rebaudiana* plants without the inoculation of endophytic fungi, i.e., control group. (**B**) Plants after inoculation with endophytic fungi (*F. fujikuroi*).

**Table 1 plants-12-01151-t001:** Antagonistic activity of fungal isolates from *O. tenuiflorum* against pathogenic fungi (*F. oxysporum* and *R. necatrix*).

Pathogenic Fungi	Isolated Endophytic Fungi	Percentage of Growth Inhibition
*Fusarium oxysporum*	R1	38.09 ± 2.72
	R2	45.28 ± 2.72
	R15	32.13 ± 3.09
*Rosellinia necatrix*	R11	26.98 ± 2.75
	R2	42.85 ± 2.75
	R12	28.57 ± 2.38

**Table 2 plants-12-01151-t002:** UHPLC analysis of polyphenols from different extracts of *S. rebaudiana*.

Peak Name	Methanolic Extract of Inoculated Plant			Methanolic Plant Extract (Control)			Chloroform Extract of Inoculated Plant			Chloroform Plant Extract (Control)		
	Retention time Min	Relative Area %	Amount (mg/L)	Retention Time Min	Relative Area %	Amount (mg/L)	Retention Time Min	Relative Area %	Amount (mg/L)	Retention Time Min	Relative Area %	Amount (mg/L)
Gallic acid	2.41	1.78	0.04	2.44	1.52	0.01	2.42	5.23	0.06	2.28	0.11	0.08
Caffeic acid	3.08	2.86	0.55	3.07	5.22	0.46	3.06	7.38	0.18	3.06	6.41	0.10
Syringic acid	3.27	44.01	5.44	3.25	54.22	3.06	3.33	2.24	1.59	3.31	0.48	0.34
Rutin	8.593	93.08	20.8793	8.567	100	3.2362	8.707	100.00	5.1032	8.713	100.00	0.0818
Quercetin	14.700	2.42	N.D.	N.D.	N.D.	N.D.	N.D.	N.D.	N.D.	N.D.	N.D.	N.D.
p-Coumaric acid	N.D.	N.D.	N.D.	N.D.	N.D.	N.D.	N.D.	N.D.	N.D.	N.D.	N.D.	N.D.
Salicylic acid	N.D.	N.D.	N.D.	N.D.	N.D.	N.D.	N.D.	N.D.	N.D.	N.D.	N.D.	N.D.
Total		100.00			100.00		1.855	100.00			100.00	

N.D.—Not Determined.

**Table 3 plants-12-01151-t003:** Primers used for amplification of sequences.

S. No.	Oligo Name	Sequence (5`à 3`)	Tm (°C)	GC Content
1.	ITS Forward	TCCGTAGGTGAACCTGCGG	57	63.15%
2.	ITS Reverse	TCCTCCGCTTATTGATATGC	53	45%

## Data Availability

The datasets used and/or analyzed during the current study are available from the corresponding author on reasonable request.

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
