# Peer review of "Improvement in the Phytochemical Content and Biological Properties of Stevia rebaudiana (Bertoni) Bertoni Plant Using Endophytic Fungi Fusarium fujikuroi"

_plants, 2023, doi:10.3390/plants12051151_

Round 1

Reviewer 1 Report (Previous Reviewer 1)

The manuscript "Sustainable improvement in the phytochemical content and biological properties of Stevia rebaudiana plant inoculation with endophytic fungi." (plants-2194464) describes the isolation of twenty endophytic
fungi from the medicinal plant Ocimum tenuiforum and the use
of one of them (isolate R2, identified as Fusarium fujikuroi
isolate R2 OS) to improve the phytochemical content and
biological properties of Stevia rebaudiana, another medicinal
plant.

This manuscript is a second version of the manuscript with the same name (but with the reference plants-2070250) that I evaluated in November 2022. In this new version all the changes I suggested have been resolved, except for the change from nigrican to nigricans (line 164 of the current manuscript). The references are not homogeneous either, at least in terms of how to cite the names of the journals. I would have no objection to accepting the article in its current form once the mentioned errors are corrected.

Author Response

Reviewer 2 Report (Previous Reviewer 4)

This article studied antifungal and biological effect of endophytic fungal strains against phytopathogens. The study presents some valuable findings in the field of plant pathology. This study can facilitate further role of plant extracts and its different biological effects on plants. Before recommending this article for publication, there are some shortcomings for that should be resolve.

This work is much improved from the previous version. However still there are many deficiencies which needs to be improve.

The title of the work used word sustainable and also claim that endophytic bacteria improve its phytochemical contents. However, the experiments performed in this study are not enough for this title. So the authors should change the title.

Line 36 “twenty endophytic fungigal were isolated” add word fungal strains

FRAP assay use full form at first use.

Molecular characterization of only one isolate was performed in this study what about others?.

 Line 67 “In return, EF” use consistent terms or abbreviations throughout the MS.

What is the novelty in this study. If previous studies have proved.

Also write the aims and objectives of this study clearly in the introduction section.

Section 2.2 only one strain was identified through molecular study. How the authors can name other strains only by morphological properties.

Section 3.2 and 3.6 could be cited with recent studies,  https://doi.org/10.1016/j.bcab.2020.101729, https://doi.org/10.1007/s10534-022-00417-1 respectively.

Write full form of UHPLC at first use.

Discussion and conclusions are well modified.

Author Response

Reviewer 3 Report (New Reviewer)

This study aims to enhance the medicinal potential of the plant through endophytic fungi inoculation, the results may contribute to medicinal plants for increasing their phytochemical content and hence medicinal potential. However, the writing is bad and the organization is disordered. Additional questions and suggestions are as following.

1.  The abstract section is not concise or accurate, too much background information has been provided.  rutin (a flavonoid) and syringic acid (a polyphenol) were identified? If you like to use brackets, rutin (flavonoid) and syringic acid (phenolic acid) may be better. phenolic compounds or polyphenols,  mainly include phenolic acids, flavonoids,etc., please check in the other part of the whole manuscript, for example, line 247.

2. ppm should be replaced by standard units, please check in the whole manuscript.

3. Introduction section could be summarized.

4. Key words section, did not provide the optimal key words and delete the some unessential ones.

5. The structure of this manuscript should be reorganized. section 2.1 Statistical analysis should be included in material and methods, not in results section. And some methods and results are mixed.  

6. UPLC chromatograms are not clear.

7. So many spelling mistakes should be checked.

This study aims to enhance the medicinal potential of the plant through endophytic fungi inoculation, the results may contribute to medicinal plants for increasing their phytochemical content and hence medicinal potential. However, the writing is bad and the organization is disordered. Additional questions and suggestions are as following.

1.  The abstract section is not concise or accurate, too much background information has been provided.  rutin (a flavonoid) and syringic acid (a polyphenol) were identified? If you like to use brackets, rutin (flavonoid) and syringic acid (phenolic acid) may be better. phenolic compounds or polyphenols,  mainly include phenolic acids, flavonoids,etc., please check in the other part of the whole manuscript, for example, line 247.

2. ppm should be replaced by standard units, please check in the whole manuscript.

3. Introduction section could be summarized.

4. Key words section, did not provide the optimal key words and delete the some unessential ones.

5. The structure of this manuscript should be reorganized. section 2.1 Statistical analysis should be included in material and methods, not in results section. And some methods and results are mixed.  

6. UPLC chromatograms are not clear.

7. So many spelling mistakes should be checked.

Round 2

Reviewer 3 Report (New Reviewer)

Thanks for improving the manuscript. This is absolutely a better manuscript than the previous one. However, major revisions still need to be done. Based on your response to my comments, the responses on comment 1,6 and 7 are not enough. I would be happy if you revised it based on these comments.

1.    If you like to use brackets, rutin (flavonoid) and syringic acid (phenolic acid) may be better. phenolic compounds or polyphenols, mainly include phenolic acids, flavonoids, etc., please check in the other part of the whole manuscript.

2.    UPLC chromatograms are not qualified and these chromatograms can not be published, chromatograms of the samples maybe need to be rerun.

3.    So many spelling mistakes should be checked, such as mg/L,  μg/mL.

Author Response

Response Sheet

Thanks for the valuable suggestions. The corrections have been done in the main manuscript file as desired.         

  1. If you like to use brackets, rutin (flavonoid) and syringic acid (phenolic acid) may be better. phenolic compounds or polyphenols, mainly include phenolic acids, flavonoids, etc., please check in the other part of the whole manuscript.

Response: Corrections have been done in the whole manuscript i.e Rutin, quercetin, gallic acid, caffeic acid and syringic acid (polyphenols) present in the plant extracts (methanol and chloroform) might be responsible for different bioactivities which were assessed for their presence in plant extracts by UHPLC analysis.

  1. So many spelling mistakes should be checked, such as mg/L, μg/mL.

Response: Spelling mistakes have been corrected.

  1. UPLC chromatograms are not qualified and these chromatograms can not be published, chromatograms of the samples maybe need to be rerun.

Response: After processing the plant samples once more, new chromatogram have been included to the manuscript.

Round 3

Reviewer 3 Report (New Reviewer)

Extensive revisions have been done, could be accepted.

This manuscript is a resubmission of an earlier submission. The following is a list of the peer review reports and author responses from that submission.

Round 1

Reviewer 1 Report

The manuscript "Sustainable improvement in the phytochemical content and biological properties of Stevia rebaudiana plant inoculation with endophytic fungi." (plants-2070250) describes the isolation of twenty endophytic fungi from the medicinal plant Ocimum tenuiforum and the use of one of them (isolate R2, identified as Fusarium fujikuroi isolate R2 OS) to improve the phytochemical content and biological properties of Stevia rebaudiana, another medicinal plant.

The work is technically well done, although it does not present much originality: despite what is mentioned at the end of the abstract “This approach can further be utilized for other medicinal plants for increasing their phytochemical content and hence medicinal potential in a sustainable way”, there are many articles that have already used this approach to improve the therapeutic potential of other medicinal plants. I cite three examples found in a quick search (not included in the references of this manuscript), to which should be added that are indeed included and are similar approachs (refs. 33 to 36).

Gupta, S; Chaturvedi, P

Enhancing secondary metabolite production in medicinal plants using endophytic elicitors: a case study of Centella asiatica (Apiaceae) and asiaticoside

IN: ENDOPHYTES FOR A GROWING WORLD. PART IV - ENDOPHYTES FOR NOVEL BIOMOLECULES AND IN VITRO METHODS (CHAPTER 14, PP. 310-327). HODKINSON, TR; DOOHAN, FM; SAUNDERS, MJ; MURPHY, BR (EDS.), DOI: 10.1017/9781108607667, CAMBRIDGE UNIVERSITY PRESS, CAMBRIDGE, UNITED KINGDOM. APR 2019

Ye, HT; Luo, SQ; Yang, ZN; Wang, YS; Ding, Q; Wang, KF; Yang, SX; Wang, Y Endophytic fungi stimulate the concentration of medicinal secondary metabolites in Houttuynia cordata Thunb.

PLANT SIGNALING & BEHAVIOR. 16 (9): 1929731, DOI: 10.1080/15592324.2021.1929731; SEP 2021

Jia, M; Chen, L; Xin, H-L; Zheng, C-J; Rahman, K; Han, T; Qin, L-P

A friendly relationship between endophytic fungi and medicinal plants: a systematic review

FRONTIERS IN MICROBIOLOGY, 7: 906; DOI: 10.3389/FMICB.2016.00906, JUN 2016

The manuscript needs a thorough review of the English language, and, as I will mention later, several of the figures are unnecessary, and the description of the molecular identification methodology is inconsistent.

For these reasons, I consider that in its current state, the manuscript should be rejected.

Some other comments to improve the manuscript:

- Line 43. Keywords must be placed, preferentially, in alphabetical order.

- Line 60. The Genus name of a biological name must be completed the first time it is cited (e.g., F. thapsinum. F. proliferatum (both in line 60), but also F. verticillioides (line 66), O. tenuiflorum, (line 90), R. necatrix (line 117), and F. oxysporum (line 117).

- Line 101. Figures 1 and 2 are no needed, because only one of the isolates were used in the work. The two figures do not add anything additional to the work. Additionally, the legend of figure 1 does not have italics in any of its scientific names, and there are up to five errors in genus or species names: Penicillium (twice, line 105), Epicoccum (also in line 105), ramosa (line 106), and nigricans (line 107).

- Line 109. Already mentioned in the previous point. There are also five errors in genus or species names: Penicillium (twice, line 112), Epicoccum (also in line 112), ramosa (line 113), and nigricans (line 114).

- Line 124. Figure 3 is superfluous since the numerical interpretation is included in Table 1. The paragraph between lines 127 and 136 would have to be redrafted.

- Lines 130-131. “against both pathogenic fungi was 42.82%, 45.82%”. The “and” between the two numbers is missing.

- Lin 165. Figure 4 is not referenced in the text...

- Line 175. Quantities and units must be separated (e.g., 100 μg/ml). This also applies to, at least, lines 179, 180, 181, 273, 314, 321, 322, 323, 324, 370, 371, 372, and 421.

- Line 181. Fig.3 must be separated as Fig, 3. But the text is not really mentioning figure 3 but rather figure 5...

- Line 193. Milliliter has been written up to this point as ml, and two ways of writing it can't be mixed, so mL must be changed to ml. This also applies to lines 194 and 196 (twice).

- Line 232. “inoculated S.rebaudiana”. S. rebaudiana must be separated. This also applies to lines 243.

- Line 245. Figure 9 needs a dot, after 9. S. rebaudiana lacks italics (twice).

- Line 249. Figure 10 needs a dot, after 10. S. rebaudiana lacks italics (twice).

- Line 259. “plant sample”. Please, change to samples, it is plural.

- Line 310. “0.1 volume of 3 M Sodium acetate”. Sodium has an unnecessary capital letter.

- Line 319: It is not clear what the RAPD was or would be used for

- Line 323: “and 1 l (5U/l) of dNTPs were used in the PCR reactions”. One liter???

- Lines 316-328. The PCR method for ITS amplification is not described, nor are the primers used.

- Line 333. The phylogenetic tree is not shown, the identification could have been made perfectly by using BLAST.

- Line 342. Erlenmeyer, please capitalize.

- Line 343. 120 rpm/min. rpm already includes the time variable

- Line 409. “[31].On the other hand”. Missing space after dot.

- Line 419. “under heat stress respectively”. Missing comma before respectively.

- Line 476. “and writing manuscript RY, SM and AK”. Missing comma before RY.

Reviewer 2 Report

It was a pleasure to review the manuscript entitled “Sustainable improvement in the phytochemical content and biological properties of Stevia rebaudiana plant inoculation with endophytic fungi" written by Reema Devi et al.

Authors highlight that the fungal isolated from the medicinal plant Ocimum tenuiflorum have shown antagonistic activity towards plant pathogenic fungi R. necatrix and F. oxysporum, and the extracts of the S. rebaudiana plant had shown higher antioxidant potential when inoculated with the endophytic fungi as compared with control plant. The aim of the study is clear and well-conducted but lack of reasonable analysis and discussion, so it needs extensive revision before publication.

I agreed with that the endophytes are endosymbionts that asymptomatically coexist with plants for least part of their lives without causing disease. So the fungal strains (R1-R20) isolated from Ocimum tenuiforum only temporarily do not cause the plant to show symptoms of disease. It did not mean that these strains are not exactly pathogenic. In this study, the authors conducted dual culture experiment to investigate the antagonistic effect of the fungal strains and conducted subsequent DPPH and FRAP experiments using the most antagonistic strains, while the pathogenicity tests were not done. Therefore, the endophytic fungal used in the title actually refer to the strains with antagonistic effect, rather than all endophytic fungi, so it would be more appropriate to use the species name directly.

Most of the fungal stains isolated from O. tenuiforum identified using morphological characteristics and the R2 was identified combined with phylogenetic analysis based on ITS sequence. Finally, the authors showed the identification results of several strains at species level. In general, accurate identification of fungal strains at the species level requires a combination of morphological characteristics and multilocus sequence typing based on ITS and other house-keeping genes. So, I am not sure if the species identification results in this study were accurate. Additionally, the pictures showing the fungal colony morphology (Fig. 1) were not very clear. Moreover, it does not seem to be a pure culture colony morphology, for example, R8,R14,R15,R19, etc. Were they obtained by monospore culturing? The Fig.2 were not very clear, either. And please indicate in the note what exactly is shown in the pictures.

According to Fig3, R2 did not seem to have a significant inhibitory effect on R. necatrix, and no inhibition zone was formed.

In Table1, percentage of growth inhibition should be mean± standard deviation based on multiple biological replicates.

There's no 16S rRNA sequence in fungus, so I think that was a mistake on line 94 and 142.

Line 161: please show the phylogenetic tree.

There was no explanation of Fig. 4.

Line 181: I do not think the Fig.3 can depict this. It should be Fig. 5.

Line256-257: Please unify the writing of geographical coordinates.

Line 367: please remove the “.” of  “2,2-. Diphenyl”.

Line 376: The left parenthesis is missing in the formula.

Reviewer 3 Report

The manuscript "Sustainable improvement in the phytochemical content and biological properties of Stevia rebaudiana plant inoculation with endophytic fungi" should report data on the isolation of fungal endophytes, their antifungal activity and their role in enhancing Stevia antioxidants biosynthesis. The topic is interesting, but the whole text is hard to evaluate due to many problems:

- the English language level is low, and should be improved; moreover, the text contains many repeated sentences

- the Abstract and Introduction are not properly constructed, as the former reports a mix of methods and unnecessary information, lacking the needed synthesis, and the latter is a collection of mixed sentences and references without a logical thread, not useful to the reader to understand the basis on which the authors built their work

- the results section follow the same style, often repeating the same information, or inserting unnecessary findingsw (e.g.the R2 sequence) with a general lack of useful findings: no phylogenetic tree is reported, no data variation and no results of statistic analyses are reported in text and/or tables, and the post-Hoch derived lettering of figures 5 and 6 is not correct

- colours in figures are hardly visible and should be changed

- the discussion is constructed in analogy with the other sections, thus it is not properly discussing the findings of this work

Reviewer 4 Report

This article studied role of endophytic fungi in improvement of phytochemicals and biological properties Stevia rebaudiana. This study can facilitate further role of plant extracts and its different biological effects on plants. Before recommending this article for publication, there are some shortcomings for that should be resolve.

Write complete accepted name of the species in the title.

Remove full stop at the end of the title.

Use full form of the abbreviations at first use.

Line 54 should be cited with relevant study as the authors stated about research studies here. The following studies could be possibly cited here.

https://doi.org/10.1007/s10534-022-00417-1, https://doi.org/10.1016/j.pmpp.2021.101639,

Line 78 specify the secondary metabolites.

Is there any direct evidence or mechanism that fungi play role in the enhancement of medicinal values of the plants?

The authors have presented numerous studies reviewing the potential of Endophytic fungi but no direct evidence has been presented.

The objective of this study need to be revise and the title as well because there several factors effecting plant phytochemicals at one time.

Add economic and medicinal importance of the studied plants in the introduction.

Italicize the species names in the whole MS specifically in figure legends.

Table 1 why only few are shown.

Section 2.4 where is the phylogenetic tree.

Check position of figure 4. What its representing and where is its description.

Discussion must be made with relevant and recent studies.

Conclusion looks like summary some points should be added like what these results conclude and what are its future impacts.